# Utilization of Stem Cells in Medicine: A Narrative Review

**DOI:** 10.3390/ijms26199659

**Published:** 2025-10-03

**Authors:** Banu Ismail Mendi, Rahim Hirani, Alyssa Sayegh, Mariah Hassan, Lauren Fleshner, Banu Farabi, Mehmet Fatih Atak, Bijan Safai

**Affiliations:** 1Department of Dermatology, Nigde Omer Halis Demir University, Niğde 51000, Türkiye; banuismail92@gmail.com; 2School of Medicine, New York Medical College, Valhalla, NY 10595, USA; rhirani2@student.nymc.edu (R.H.); asayegh4@student.nymc.edu (A.S.); mhassan7@student.touro.edu (M.H.); fatih9164@hotmail.com (M.F.A.); 3Department of Dermatology, Icahn School of Medicine at Mount Sinai, New York, NY 10029, USA; 4Department of Dermatology, NYC Health + Hospitals/Metropolitan, New York, NY 10029, USA; 5Department of Dermatology, NYC Health + Hospitals/South Brooklyn Health, New York, NY 11235, USA

**Keywords:** stem cells, regenerative medicine, embryonic tissue, cell therapy, pluripotency

## Abstract

Regenerative medicine holds significant promise for addressing diseases and irreversible damage that are challenging to treat with conventional methods, making it a prominent research focus in modern medicine. Research on stem cells, a key area within regenerative medicine due to their self-renewal capabilities, is expanding, positioning them as a novel therapeutic option. Stem cells, utilized in various treatments, are categorized based on their differentiation potential and the source tissue. The term ‘stem cell’ encompasses a broad spectrum of cells, which can be derived from embryonic tissues, adult tissues, or generated by reprogramming differentiated cells. These cells, applied across numerous medical disciplines including cardiovascular, neurological, and hematological disorders, as well as wound healing, demonstrate varying therapeutic applications based on their differentiation capacities, each presenting unique advantages and limitations. Nevertheless, the existing literature lacks a comprehensive synthesis examining stem cell therapy and its cellular subtypes across different medical specialties. This review addresses this lacuna by collectively categorizing contemporary stem cell research according to medical specialty and stem cell classification, offering an exhaustive analysis of their respective benefits and constraints, thereby elucidating multifaceted perspectives on the clinical implementation of this therapeutic modality.

## 1. Introduction

Stem cells have recently gained prominence as a promising therapeutic approach. These unspecialized cells possess the remarkable ability to differentiate into any specific cell type within an organism. This characteristic enables them to self-renew and be utilized in the treatment of various diseases. Stem cells are present throughout all stages of life, including adulthood, with a high concentration in embryos. However, their differentiation potential diminishes over time, limiting their ability to transform into specific cell types within certain organs. Despite this temporal reduction in differentiation potential, each developmental stage at which stem cells undergo differentiation presents distinct advantages and limitations that vary according to the specific disease context [1,2]. Although stem cell therapy has gained increasing traction across numerous medical disciplines, the literature lacks comprehensive studies that collectively compile the applications of stem cell therapy across different specialties. This review seeks to provide a unified evaluation of stem cell therapy utilization across various medical fields, examining both its benefits and limitations from an integrated perspective.

The study selection process involved a comprehensive search of PubMed and Web of Science databases for literature published between 1990 and 2025. The search strategy utilized the terms “stem cell” and “stem cell therapy” in conjunction with discipline-specific terminology corresponding to each relevant medical specialty, including dermatology, nephrology, aesthetic medicine, endocrinology, neurology, cardiology, hematology, gastrointestinal and musculoskeletal diseases, infertility and oncology.

## 2. Classification of Stem Cells

Stem cells are categorized based on their differentiation potential, ranging from highest to lowest potency, as totipotent, pluripotent, multipotent, oligopotent, and unipotent [3] (Figure 1).

### 2.1. Totipotent Stem Cells

Totipotent cells represent the stem cells with the highest differentiation potential, capable of giving rise to both the embryo and extra-embryonic structures. A prime example of a totipotent cell is the zygote, which forms immediately following fertilization [1].

### 2.2. Pluripotent Stem Cells

During the blastocyst stage, two distinct cell groups are present: the trophectoderm and the inner cell mass. The trophectoderm develops into the placenta, while the stem cells within the inner cell mass, which are pluripotent, form the fetus. Pluripotent stem cells have the ability to differentiate into endoderm, ectoderm and mesoderm cells. Unlike totipotent stem cells, these pluripotent cells do not give rise to extraembryonic structures. Embryonic stem cells are derived from the pluripotent cells of the inner cell mass [1,2].

### 2.3. Multipotent Stem Cells

Pluripotent cells differentiate into multipotent stem cells following the formation of germ layers. The differentiation capacity of multipotent stem cells is more restricted compared to pluripotent stem cells, as they can only generate cell types within their specific germ layer. Hematopoietic stem cells and mesenchymal stem cells serve as examples of multipotent stem cells [1].

### 2.4. Oligopotent Stem Cells

Multipotent cells further differentiate into oligopotent cells. As the name implies, oligopotent cells have a more restricted differentiation capacity, capable of developing into only a few cell types. Myeloid progenitor cells are an example of oligopotent cells [1,4].

### 2.5. Unipotent Stem Cells

Unipotent stem cells can differentiate into a single specific cell type, possessing the most limited differentiation capacity. However, their ability to repeatedly divide underscores their potential use in regenerative medicine [1].

### 2.6. Induced Pluripotent Stem Cells

Differentiated cells can be reprogrammed to generate pluripotent stem cells, known as induced pluripotent stem cells. These cells possess properties similar to embryonic stem cells and are used in drug development, disease modeling, and regenerative therapy [2]. Table 1 presents a comparison of various developmental stages of stem cells.

## 3. Therapeutic Applications of Stem Cells

### 3.1. Cardiovascular Diseases

Cardiovascular diseases are the leading cause of mortality and morbidity worldwide. Despite recent advances in treatment, the prognosis remains poor, and patients’ quality of life is significantly diminished due to the heart cells’ limited regenerative capacity. Stem cell therapy offers promising potential to address this gap with recent advancements [15,16]. The use of stem cells in cardiovascular diseases aims to reduce heart damage and improve cardiac function, with a focus on ischemic diseases and heart failure. This section will discuss the studies conducted in this area.

Recently, cardiac stem cells (CSCs) with regenerative potential have been identified, but their numbers decline with age [17]. Although animal studies have yielded conflicting results regarding CSCs [18,19,20], phase 1 and phase 2 clinical trials with a limited number of patients have reported that CSCs reduce infarct size, increase cardiac contractility, and ultimately improve symptoms [21,22,23,24,25]. Due to the limited availability of these cells, skeletal myoblasts, derived from satellite cells and involved in the repair of skeletal muscle cells, have been used in heart regeneration. These cells are advantageous due to their autologous nature, ease of acquisition, ability to tolerate ischemic conditions, myogenic capacity, low tumorigenic potential, and rapid expansion ability [26,27,28]. Although preclinical studies with these cells reported improvements in myocardial performance and reduced remodeling [29,30,31,32], their use was reduced due to severe arrhythmogenic effects observed in subsequent clinical studies [27,33].

Due to the extensive differentiation capabilities of embryonic stem cells, research on cardiac regeneration has been conducted both in vitro and in vivo [15]. These studies have demonstrated that embryonic stem cells can differentiate into cardiomyogenic cells in suitable environments, positively influence cardiac remodeling through paracrine and autocrine effects, reduce fibrosis, and exhibit anti-apoptotic properties [16]. In a 2018 clinical study, cardiac progenitors derived from embryonic stem cells were administered to patients with heart failure, resulting in improvements in systolic function and a reduction in symptoms [34]. Although human embryonic stem cells show promise, their use is limited by ethical concerns, the risk of teratoma development, potential rejection, and the necessity of immunosuppressive drugs to prevent rejection [16,35].

Induced pluripotent stem cells (iPSCs) offer significant advantages over embryonic stem cells, as they circumvent ethical issues and can be generated from adult cells, providing an unlimited resource [15]. Cardiomyocytes derived from iPSCs have been shown to exhibit anti-apoptotic effects, reduce fibrosis, and increase ejection fraction in animal models [36,37,38]. However, recognizing that many iPSCs do not survive post-transplantation in ischemic environments, researchers transplanted microvessels derived from adipose tissue alongside iPSCs in a mouse model, which enhanced the survival of cardiomyocytes and the maturation of iPSCs into cardiomyocytes [39]. Although large-scale human studies are lacking, in 2023, a cardiomyocyte patch derived from iPSCs was applied to three patients with ischemic cardiomyopathy, resulting in alleviated heart failure symptoms and improved heart contractility [40]. iPSCs are also utilized in disease modeling and drug screening [41]. Despite their advantages, the proto-oncogenes used during reprogramming and mutations from the original cell state can lead to genomic instability and tumorigenic properties in iPSCs. To mitigate these issues, recent approaches have employed non-integrated or non-viral gene delivery vectors, recombinant proteins, and microRNAs in the generation of iPSCs [15,16].

Mesenchymal stem cells (MSCs) can also be used in the treatment of cardiovascular diseases, as they are capable of differentiating into cardiomyocytes when cultured with cardiomyocytes or exposed to agents such as 5-azacytidine and retinoic acid [16]. Lacking hematopoietic and endothelial surface proteins, MSCs can evade immune surveillance, making them suitable for both autologous and allogeneic transplantation. They also possess immunomodulatory and anti-inflammatory properties [35]. Their effectiveness is primarily due to their paracrine effects rather than direct differentiation. Recent studies have demonstrated their role in cardioprotection by enhancing angiogenesis, promoting proliferation, and reducing apoptosis through the exosomes they secrete [42]. Endothelial progenitor cells (EPCs) are another type of stem cell used for cardiovascular diseases, given that some of these conditions are caused by endothelial dysfunction [43,44]. EPCs migrate to damaged areas to stimulate proliferation through paracrine effects. Early phase 1 and phase 2 studies reported that EPCs reduced infarct size [45,46,47,48]; however, no significant improvement in cardiac function was observed in long-term follow-up [49,50]. Similar results were reported in subsequent phase 3 studies [51,52,53]. Table 2 demonstrates sample clinical studies about stem cell utilization in cardiovascular diseases.

### 3.2. Dermatology

#### 3.2.1. Psoriasis

Psoriasis lesions have been observed to improve and enter long-term remission in patients undergoing hematopoietic stem cell therapy [59]. Conversely, psoriasis lesions have appeared in recipients who received bone marrow from donors with psoriasis [60,61]. Recognizing this, a considerable number of reports have documented psoriasis remission following hematopoietic stem cell transplantation—whether allogeneic or autologous—prompting speculation that stem cells may play a role in the treatment of psoriasis [62]. In allogeneic therapy, remission can last up to 20 years, whereas relapse is more frequent in autologous therapy [63].

In preclinical studies, umbilical cord mesenchymal stem cells (UCMSCs) have been shown to regulate Th1 and Th17 cells, which are crucial in the pathogenesis of psoriasis [64]. Clinical studies have demonstrated significant improvement in patients receiving UCMSCs, with no serious side effects reported [65,66]. UCMSCs have advantages over other MSCs, as they are derived from fetuses, exhibit lower immunogenicity, and possess stronger therapeutic effects compared to adult MSCs [64]. Adipose-derived mesenchymal stem cells (ADMSCs) have also been found to reduce inflammation by upregulating TGF-β and inhibiting keratinocyte proliferation [67]. Clinical studies and case reports have shown that ADMSC treatment results in regression of psoriasis lesions without serious side effects [68,69]. While adipose MSCs have been successful, their efficacy may be limited by the differences in cytokine and antigen secretion, as well as the distinct microenvironment of stem cells in the skin and bone marrow compared to healthy controls [70].

#### 3.2.2. Atopic Dermatitis

Preclinical studies have indicated that AD-MSCs can alleviate inflammation by reducing IgE levels, degranulated mast cells, histamine, prostaglandin E2, and proinflammatory cytokine secretion [71]. In a clinical study, four patients with atopic dermatitis (AD) resistant to conventional treatment were successfully treated with autologous AD-MSCs [72]. However, the autologous application of AD-MSCs in AD patients is questionable, as abnormal Th1/Th17 cytokine and chemokine levels, thought to be involved in the disease’s pathogenesis, were found in MSCs obtained from skin samples of these individuals [73]. Human umbilical cord blood stem cells (UCB-MSCs) have also been found effective in treating AD. In a clinical study involving 34 patients with moderate to severe AD, high-dose human umbilical cord blood-derived MSC application significantly improved lesions without side effects [74]. Additionally, bone marrow stem cells (BMSCs) have shown therapeutic effects on AD by reducing serum IgE levels and decreasing cell infiltration in skin lesions [75,76,77].

#### 3.2.3. Vitiligo

The popularity of cell and tissue grafts in treating vitiligo is increasing. Recently, the autologous non-cultured outer root sheath hair follicle cell suspension (NCORSHFS) technique, which leverages the regenerative capacity of hair follicle melanocytes, has been reported to achieve an average of 65.7% repigmentation in patients [78]. Another cell type investigated in this context, multi-lineage differentiation stress enduring (MUSE) cells, which are pluripotent stem cells, have been shown to differentiate into melanocytes [79]. Additionally, ADSCs have the ability to differentiate into melanocytes and, when co-cultured with melanocytes, can enhance melanocyte function [80,81]. Meanwhile, dermal MSCs inhibit the proliferation of CD8+ lymphocytes, which play a crucial role in the pathogenesis of vitiligo [82].

#### 3.2.4. Epidermolysis Bullosa

Various types of stem cells have been investigated for the treatment of epidermolysis bullosa. Although positive outcomes, such as decreased inflammation and accelerated wound healing, have been observed with HSCs, research in this area has recently declined due to the risk of death from graft-versus-host disease and myeloablative conditions. Instead, there is increasing interest in MSCs, which can reduce systemic inflammation and promote local wound healing when administered systemically, and are considered safer. While both types have shown clinical success, UCB-MSCs are reported to have higher proliferative, regenerative, and immunosuppressive capacities with less immunogenicity than BM-MSCs. Recently identified in human dermis, ABCB5+ MSCs with immunomodulatory properties have also been found effective in recessive dystrophic epidermolysis bullosa (RDEB) in preclinical and clinical studies. This suggests that the long-term systemic use of these cells may improve the structural integrity of the skin and mucous membranes due to their high adhesion properties to damaged tissues (compared to BM-MSCs) and their ability to secrete collagen VII [83].

#### 3.2.5. Alopecia

Stem cell applications have been utilized in treating androgenetic alopecia and alopecia areata. Gentile et al. injected human adult stem cells, isolated by centrifuging human hair follicles obtained via punch biopsy, into the scalps of 11 patients with androgenetic alopecia (AGA) [84]. This resulted in an increase in hair density and number compared to the initial state and placebo. Similar positive outcomes were observed in subsequent clinical studies [85,86]. Additionally, fat-derived stem cell conditioned medium, administered by injection into the scalp or through microneedling, proved effective in treating AGA [87,88]. A new stem cell method, known as “stem cell educator therapy,” has also been introduced for treating alopecia areata. In this approach, the patient’s mononuclear cells are separated from whole blood and allowed to interact with multipotent stem cells obtained from human cord blood. These “educated” cells are then returned to the patient’s circulation, yielding promising results in patients with severe alopecia areata [89].

#### 3.2.6. Systemic Sclerosis

HSCT has been extensively studied for systemic autoimmune diseases such as systemic sclerosis (SSc) and systemic lupus erythematosus. This therapy has been successful in both conditions, but it is particularly recommended for resistant cases. Specifically, for SSc, HSCT is advised for patients with resistant, acute onset, rapid progression, and mild initial organ damage. Studies suggest that these patients have a better prognosis following treatment. However, long-term disease, a slow progression, and irreversible organ damage are considered contraindications for this therapy [90]. MSCs has also been explored in preclinical and clinical studies for SSc [91,92,93]. It has been shown to reduce skin necrosis by improving blood flow and vascularization in SSc patients [94,95].

Table 3 presents sample clinical studies categorized by dermatological diseases and stem cell types.

### 3.3. Plastic Surgery and Aesthetic Medicine

#### 3.3.1. Wound Healing

Studies have shown that when MSCs are administered to wound areas, they enhance fibroblast migration and angiogenesis, stimulate extracellular matrix deposition, facilitate wound closure, initiate re-epithelialization, and exhibit immunomodulatory effects [108]. Successful outcomes have been achieved with BM-MSCs, ADSCs, MSCs from the placenta, and the newly discovered ABCB5+ mesenchymal cells [109,110]. In one study, MSCs from bone marrow were applied to non-healing wounds using a fibrin spray system, yielding successful results [111]. In a double-blind randomized controlled study, BM-MSCs were administered to patients with diabetic ulcers, achieving complete healing within four weeks [112]. ADSCs have also proven successful in wound healing in both animal and human studies, primarily due to their paracrine effects rather than direct cell replacement. The ADSC secretome contains numerous growth factors that promote epithelialization and angiogenesis [113].

#### 3.3.2. Scars

Research has demonstrated that MSCs can reduce scar formation and existing scars. In animal models, UC-MSCs and BM-MSCs have been reported to reduce hypertrophic scar development by inhibiting the transformation from myofibroblasts to fibroblasts, reducing fibroblast proliferation, and controlling inflammation [114,115,116,117]. Additionally, ADSCs have been shown in both in vivo and in vitro models to reduce fibroblast proliferation and migration, as well as collagen deposition, thereby preventing hypertrophic scar and keloid development [118]. Since ADSCs can be easily collected through procedures such as liposuction, they are frequently used in scar treatment and can be applied using techniques like fat grafting or stromal vascular fraction [119]. Furthermore, a study comparing the effectiveness of BM-MSCs and ADSCs noted that both were more effective in reducing TNF-alpha and IL1-beta, which are important in scar development, and in increasing the MMP1/TIMP1 ratio [120]. Regarding MSC conditioned media, clinical studies have demonstrated that combining it with laser therapy results in significantly greater improvement in the appearance of post-acne atrophic scars compared to laser therapy alone [121]. Another emerging area of interest is the application of exosomes derived from MSCs, which have been shown to prevent scar formation through horizontal miRNA transfer, suppressing the differentiation of fibroblasts into myofibroblasts [114,122].

#### 3.3.3. Skin Rejuvenation

Stem cells are utilized in skin rejuvenation due to their ability to enhance fibroblast development and activity, increase collagen production, and reduce inflammation [123]. ADSCs are applied directly or as fat grafts or stromal vascular fractions, addressing both these effects and the reduction in facial fat with age [109]. Clinical studies have reported that stem cells contribute to skin rejuvenation by altering the dermal pattern in patients who received stem cells or stromal vascular fractions after facelift surgery, achieving full regeneration in cases of solar elastosis [124,125]. Additionally, clinical studies have indicated that nanofat injections can reduce lower lid pigmentation and eliminate perioral wrinkles, preventing fat accumulation or transplant migration [126]. Furthermore, in clinical studies where stem cell-conditioned media are combined with methods such as microneedling or laser, it has been shown that combined treatments are more effective than single treatments in reducing wrinkles and pores [127,128,129].

### 3.4. Infertility

#### 3.4.1. Female Infertility

The interest of stem cell research in the context of fertility appears promising, though is currently limited to preclinical research and animal studies [130]. However, stem cell-based therapies from multiple sources, including but not limited to bone marrow, umbilical cord, or menstrual blood have all demonstrated potential in restoring reproductive function.

Preclinical studies of mesenchymal stem cells (MSCs) have shown capability of secreting cytokines including vascular endothelial growth factor (VEGF), hepatocyte growth factor (HGF), and insulin growth factor-1 (IGF-1), stimulating tissue regeneration, angiogenesis and inhibiting fibrosis in ovarian and endometrial tissue [131,132]. Li et al. also reported an increased number of follicles, lower follicle-stimulating hormone (FSH) levels, higher anti-Mullerian hormone levels, and higher estradiol levels following human umbilical cord MSC transplantation in perimenopausal rats, resulting in an improved ovarian reserve function. Similarly, MSCs from menstrual blood resulted in increased ovarian weight, higher estradiol levels and increased numbers of follicles in mice [133,134].

Furthermore, MSCs have been shown to improve endometrial thickness and increase the number of endometrial receptors in mouse models [135,136]. These effects may have implications in treating uterine adhesions, thin endometrium, or even Asherman syndrome [131,137].

Granulosa cells, which are imperative in female fertility, have been explored in animal models. Stem cells from amniotic fluid have demonstrated the ability to differentiate into granulosa cells and improve overall follicular health [138]. Interestingly, micro-RNA isolated from stem cells has also shown to blunt apoptosis of granulosa cells and improve survival of ovarian follicles following chemotherapy [139].

Other stem cell types have been studied, including ovarian stem cells, embryonic stem cells, and induced pluripotent stem cells. Ovarian stem cells have shown to stimulate follicular generation and improve ovarian function [140,141]. Furthermore, they have been shown to stimulate the protein kinase B (AKT) pathway, improving fertility following in vitro activation and fragmentation with subsequent grafting of ovarian tissue back into patients with primary ovarian insufficiency, even reporting a live birth following in vitro fertilization and embryo transfer [142,143,144]. Finally, induced pluripotent stem cells have generated functional oocytes in male mouse models, decreased inflammation, and generated anti-inflammatory cytokines, though are currently considered unstable and unfit for transplantation [145,146,147,148].

#### 3.4.2. Male Infertility

Most studies exploring stem cell therapies for male infertility are focused on regenerating spermatogenesis [149]. Spermatogonial stem cells, which are self-renewing in nature, have been shown to stimulate spermatogenesis and potentially produce functional spermatids [150,151]. Furthermore, MSC transplantation has been observed to improve germ cell differentiation, spermatogenesis, testicular tissue regeneration, sperm quality and increased fertility in in vitro studies [152]. In vivo studies, however, have yet to demonstrate full spermatogenesis, though direct transplantation of testicular tissue is currently being explored [153].

Finally, one animal study showed promise following autologous grafting of cryopreserved prepubertal rhesus testis tissue [154]. Fayomi et al. reported their grafts grew and successfully produced testosterone, and resulted in a live-birth of a macaque. While these studies show promise, further exploration in safety and applicability to humans is needed.

### 3.5. Endocrinology

#### 3.5.1. Diabetes Mellitus

Stem cell therapy for diabetes mellitus (DM) has shown promising results in recent clinical trials. An allogeneic stem-cell derived islet product (VX-880) successfully produced endogenous insulin in participants with type 1 diabetes who report severe hypoglycemia [155]. In fact, one participant no longer needed insulin injections following their stem cell therapy. Recent studies have evaluated using MSCs in the treatment of Type 1 DM, exploring the use of MSCs for immunomodulation and regeneration, with mixed results. For example, one study reported no significant improvement in C-peptide levels, HbA1C or insulin needs [156]. However, studies reporting autologous adipose tissue-MSCs with vitamin D and a combination of allogenic Wharton’s Jelly MSCs with autologous bone marrow-derived mononuclear cells improved HbA1C in subjects with Type 1 DM [157,158]. Studies evaluating stem cell therapy for Type 2 DM are less promising, with predominantly short-term effects and the condition impairing stem cell function [159]. However, results were more promising for those with newer-onset T2DM, defined as <10 years and no comorbid obesity [159].

#### 3.5.2. Adrenal Insufficiency

Adrenal insufficiency, or Addison’s disease, has also been studied alongside stem cell therapy. One recent study by Ruiz-Babot et al. generated in vitro functional human-induced steroidogenic cells from human pluripotent stem cells utilizing steroidogenic factor 1 [160]. They successfully reported secretion of steroid hormones in vitro, as well as demonstrated a dose-dependent adrenocorticotropic hormone (ACTH) receptor response. While this study is still far from human introduction, it remains promising for the future of human-induced steroidogenic cells for adrenal insufficiency.

#### 3.5.3. Thyroid Dysfunction

Pluripotent stem cells have shown potential for regenerating functional thyroid cells from dermal fibroblasts [161]. In this study, human induced pluripotent stem cells were differentiated in vivo via exposure to thyroid-stimulating hormone (TSH), ethacridine and activin A. These cells, following purification and terminal differentiation, were able to express thyroglobulin, thyroid peroxidase, TSH-receptor, the sodium/iodide symporter and even secreted thyroid hormone (T4). However, a key concern remains that autoimmune attacks may occur, especially in the context of autoimmune thyroiditis.

### 3.6. Ophthalmology

#### 3.6.1. Corneal Stem Cell Therapies (Limbal Stem Cell Deficiency)

Regenerative medicine in ophthalmology is advancing rapidly due to increased activity, understanding, and function of ocular stem cells. Clinical trials are currently investigating stem cell therapies for conditions such as age-related macular degeneration (AMD), inherited retinal diseases, and limbal stem cell deficiency (LSCD) [162].

Limbal stem cell deficiency is characterized by instability of the corneal epithelium and persistent epithelial defects, which can lead to chronic pain, inflammation, and progressive vision loss [163]. Limbal stem cell transplantation promotes regeneration of the corneal surface by creating a microenvironment that supports the resettlement and function of endogenous stem cells. Mechanistically, reconstruction of defective limbal epithelial crypts occurs by providing a suitable microenvironment for resettlement of endogenous stem cells [164]. Trials using ex vivo expanded limbal epithelial stem cells (LESCs) have demonstrated favorable outcomes in restoring ocular surface stability and improving visual function in the treatment of LSCD [165,166,167].

The procedure is performed through either direct transplantation of limbal tissue or transplant of in vitro expanded cells on a biological or synthetic carrier [168]. CALEC (Cultivated Autologous Limbal Epithelial Cell) therapy at Massachusetts Eye and Ear has demonstrated over 90% success in corneal surface restoration for unilateral cases of LSCD [169].

Limbal stem cells (LSCs) have been used to treat corneal diseases and injuries, but their application has been slowed by the fragility and limitation of active stem cells of the corneal epithelium as well as the potential for iatrogenic damage to the healthy eye [168]. Clinical studies have shown excellent short- and long-term results, but transplantation comes with a high risk of complications [170]. Newer approaches involving human-induced pluripotent stem cells (hiPSCs), derived from sources such as dermal fibroblasts or hair follicles, have shown potential to generate corneal epithelial cells and three-dimensional corneal organoids [171]. These advancements offer promising therapeutic alternatives for bilateral LSCD, where autologous limbal tissue is unavailable or unsuitable.

#### 3.6.2. Retinal Pigment Epithelium Disorders (Age-Related Macular Degeneration-AMD)

Retinal diseases, such as age-related macular degeneration remains a leading global cause of irreversible vision loss in aging populations [172]. Such retinal diseases are characterized by early degeneration of the retinal pigment epithelium (RPE), followed by photoreceptor loss and choroidal thinning [173]. Initial trials of allogeneic RPE transplantation into the subretinal space showed mixed results due to immune rejection and poor cell integration [174].

Newer strategies are investigating autologous RPE cell therapies, including macular translocation and RPE-choroid patch grafts. Sharma et al. developed RPE patches derived from patient-specific iPSCs and applied them on a scaffold to support long-term cell viability and function [175]. Autologous iPS cell derived RPE cell sheet transplantation have demonstrated stability and improved visual acuity compared to choroid translocation [176]. Specifically, iPS-RPE cells from an HLA homozygous donor were transplanted in the case of wet age related macular degeneration. In all cases that underwent iPS-RPE transplantation, the presence of graft cells was indicated by areas of increased pigmentation at 6 months, which became stable during the 1 year observation period [177].

The National Eye Institute has initiated a Phase I/IIa trial using iPSC-derived RPE cells seeded onto biodegradable scaffolds for geographic atrophy [178]. Additionally, early clinical trials have developed RPE monolayers or engineered patches demonstrating that those with more severe vision loss experienced greater improvements compared with control [179]. Additionally, cell-free strategies, including intravitreal secretome or exosome injections, are being explored for their neuroprotective effects and disease-modifying potential [180].

#### 3.6.3. Inherited Photoreceptor Diseases (RP and Stargardt Disease)

Stem cell therapy also holds promise for inherited retinal dystrophies such as retinitis pigmentosa (RP) and Stargardt disease. Currently, there are no FDA-approved therapies for Stargardt disease but various approaches have been proposed for treatment. Bone marrow-derived mesenchymal stem cells (BMSCs) are being evaluated for their ability to substitute for degenerated photoreceptors and release growth factors that support cell survival [181,182]. The first application of embryonic stem cell-derived RPE cell suspension transplantation for the treatment of macular disease was in the treatment of Stargardt disease, and further studies since then have shown improvements in visual acuity [182].

Intravitreal and subretinal delivery of BMSCs has shown preliminary evidence of visual acuity improvement or delayed degeneration for inherited retinal diseases [183]. For retinitis pigmentosa, the rate of best corrected visual acuity was 49% and 30% at 6 months and 12 months, respectively. For Stargardt disease, the rate of best corrected visual acuity was 60% and 55% at 6 months and 12 months, respectively. Additionally, a newly FDA-approved trial (OpCT-001) is investigating iPSC-derived photoreceptor cell transplantation in patients with primary photoreceptor diseases, including RP [184].

A summary of key preclinical and clinical studies examining the use of stem cell therapies in ophthalmologic disorders, can be found in Table 4.

### 3.7. Pulmonology

#### 3.7.1. Chronic Obstructive Pulmonary Disease (COPD)

Stem cell therapy represents a promising avenue for COPD management due to its regenerative and immunomodulatory potential. Autologous bronchial basal progenitor cell transplantation has been associated with symptom alleviation and modest improvement in pulmonary function in early-phase trials [187]. The procedure was safe and well-tolerated, and patients demonstrated promising improvement. Additionally, an open-label pilot study administering intravenous allogeneic UC-MSCs to patients with advanced COPD demonstrated reductions in acute exacerbation frequency and symptom burden. However, pulmonary function changes were not statistically significant [188]. The therapy was well-tolerated with no infusion-related toxicities or serious adverse effects, and treated patients had significantly fewer acute exacerbations over 6 months. Supporting preclinical models have shown that UC-MSCs reduce elastase-induced emphysema and lung inflammation, reinforcing their therapeutic potential [189].

These effects are believed to be mediated through paracrine signaling, modulation of macrophage polarization toward an anti-inflammatory phenotype, and attenuation of oxidative stress within the lung microenvironment [190]. Although further trials are warranted, these studies establish a foundation for the potential clinical application of MSCs in chronic lung injury.

#### 3.7.2. Idiopathic Pulmonary Fibrosis (IPF)

IPF is characterized by progressive fibrosis of the lung interstitium, leading to alveolar collapse and respiratory failure. While no stem cell therapies are currently approved, preliminary findings support their feasibility. In a Phase I trial, autologous bone marrow-derived MSCs were endobronchially administered to patients with mild-to-moderate IPF. The treatment was well tolerated, though no significant improvements in pulmonary function were observed [191].

The limited efficacy observed in early trials may reflect challenges related to senescence of autologous cells in aged or fibrotic environments, suggesting a need for optimized cell sources, dosing schedules, or allogeneic alternatives. More recently, a preclinical study utilizing autologous proximal airway basal cells demonstrated fibrotic repair and alveolar regeneration in bleomycin-injured mice [192].

#### 3.7.3. Acute Respiratory Distress Syndrome (ARDS)

ARDS is marked by alveolar-capillary barrier disruption, protein-rich pulmonary edema, and hypoxemic respiratory failure. MSCs have shown therapeutic potential through modulation of inflammatory and endothelial responses. In preclinical models, MSCs reduced alveolar permeability, neutrophil infiltration, and expression of proinflammatory cytokines, while enhancing anti-inflammatory mediators such as IL-1 receptor antagonist, TSG-6, and insulin-like growth factor [193]. These effects are largely attributed to paracrine signaling and secreted bioactive molecules that regulate host immune responses and promote epithelial repair.

Allogeneic MSCs for ARDS in the Stem Cells for ARDS Treatment (START) trial reported that there was no improvement in outcomes, but the infusion was safe and well tolerated, so larger trials may be necessary to demonstrate efficacy [194]. Additionally, MSCs from bone marrow, adipose tissue, and umbilical cord were evaluated to determine which source is most therapeutic in inflammatory lung injury. In an ARDS-induced mouse model, umbilical cord MSCs showed superior efficacy to MSCs from bone marrow or adipose tissue. These mice had higher survival, improved lung histology, and less alveolar damage [195].

#### 3.7.4. Cystic Fibrosis (CF)

CF is an autosomal recessive disorder caused by cystic fibrosis transmembrane conductance regulator (CFTR) gene mutations, leading to chronic pulmonary infection and inflammation. Gene-edited airway stem cells have restored CFTR function ex vivo, suggesting potential for autologous cell replacement therapies. Bonfield and Lazarus recently developed a bone marrow-derived allogeneic MSC product that targets chronic lung inflammation and supports epithelial repair in CF models [196]. MSCs have been shown to suppress IL-8 secretion and modulate neutrophilic inflammation [197]. Additionally, MSC-derived antimicrobial peptides such as LL-37 may reduce pathogen burden and improve airway health.

Phase I clinical data confirm that intravenous MSC infusion is safe and well tolerated in adults with CF [198]. Additionally, preclinical trials of patient-derived airway basal stem cells that were expanded from CF patients and gene-corrected showed restored function. This preclinical trial highlights the role for autologous stem cells in gene therapy for CF [187]. Despite these promising findings, challenges remain in delivering stem cells effectively to diseased and mucus-obstructed airways, as well as ensuring long-term engraftment and functional persistence within the affected pulmonary microenvironment.

A summary of key preclinical and clinical studies examining the use of stem cell therapies in pulmonary disorders, can be found in Table 5.

### 3.8. Nephrology

#### 3.8.1. Chronic Kidney Disease (CKD)

Stem cell-based therapies, especially mesenchymal stem cells, are being explored to slow CKD progression and promote renal repair. Preclinical models show that MSCs improve kidney function and reduce pathological injury [199]. Experimental models of CKD have shown that adipose-derived-MSCs decreased damage markers in renal fibrosis, including ED-1 and α-SMA. Bone marrow derived MSCs increased mesangial thickening and macrophage infiltration, further suggesting a role in CKD treatment [200]. Early clinical trials have shown that MSCs are safe and well-tolerated in both chronic and acute kidney disease, but efficacy results have been modest [201].

Autosomal dominant polycystic kidney disease (ADPKD) is the most common genetic cause of CKD. A phase I clinical trial observed the efficacy of single-infusion MSC and demonstrated no infusion-related adverse effects and renal function stabilization. Furthermore, there was no significant improvement in eGFR or kidney cyst [200]. In addition, Saad et al. conducted a study in which patients with ischemic nephropathy due to renal artery stenosis received adipose-derived MSCs. Three months post infusion, treated kidneys showed increased cortical perfusion and GFR with reduced renal hypoxia and inflammation [202].

#### 3.8.2. Refractory Systemic Lupus Erythematosus (SLE)

MSC infusions are under study for refractory SLE, often with lupus nephritis. A pilot study by Liang et al. demonstrated that a single allogeneic MSC infusion was sufficient to achieve clinical remission in one year, with significant drops in SLE Disease Activity Index, anti-dsDNA titers, and 24 h proteinuria [203]. Furthermore, other studies have supported these results with 28–32.5% of patients in complete clinical remission in 1 year, rising to nearly 50% among those observed 4 years later [204].

Other trials noted no added benefit with aggressive immunosuppression and fading effects over time [205,206]. Recently, Kamen et al. conducted a phase 1 trial of umbilical cord-derived MSCs in patients refractory to 6 months of immunosuppressive therapy [207]. However, a small cohort (83.3%; 95% CI 35.9% to 99.6%) achieved the clinical endpoint deemed efficacious in lupus. In addition, mechanistic studies revealed reductions in CD27IgD double-negative B cells, switched memory B cells, and activated naive B cells. This trend was supported by a decreased autoantibody level in specific patients, providing insight into the mediating mechanism of MSC involvement in refractory SLE.

#### 3.8.3. Kidney Transplant

In kidney transplant, MSCs as an adjunct have been safe in both autologous and third-party forms as they show promise of lowering rejection and improving graft outcomes while minimizing drug toxicity [208]. Tan et al. demonstrated how the immunosuppressive and reparative properties of MSCs translated into reduced acute rejection and opportunistic infections when compared to anti-IL-2 receptor antibody induction therapy [208]. These results were also shown with umbilical cord derived MSCs, which significantly improved both graft and recipient outcomes but there was not a significant difference between MSC and non-MSC groups [209].

A summary of key preclinical and clinical studies examining the use of stem cell therapies in nephrology disorders, can be found in Table 6.

### 3.9. Stem Cell Applications in Neurology

Stem cell-based therapies have emerged as a promising frontier in neurology, driven by the limited regenerative capacity of neural tissues and the increasing burden of neurodegenerative and neuroinflammatory disorders. Over the past two decades, advances in stem cell biology, biomaterials, and delivery mechanisms have catalyzed the exploration of various stem cell types for neurological diseases, including MSCs, neural stem cells (NSCs), HSCs, iPSCs [212]. These cells offer neuroprotective, immunomodulatory, and regenerative potential, with ongoing preclinical and clinical investigations aiming to translate laboratory success into durable therapeutic outcomes.

MSCs are among the most widely used stem cell types in neurology, owing to their ease of isolation, immunomodulatory properties, and ability to secrete neurotrophic factors [213,214,215]. Their immunomodulatory properties are particularly beneficial in treating neurological disorders, as they can modulate the immune response, reduce inflammation, and promote tissue repair [214,216]. They have shown particular promise in the treatment of amyotrophic lateral sclerosis (ALS), where they have been associated with delayed disease progression and improved motor function in early-phase trials [217]. In a phase 1/2 and 2a clinical trial, MSCs induced to secrete neurotrophic factors (MSC-NTF) were administered intramuscularly and intrathecally to ALS patients. The treatment was found to be safe and well-tolerated, with most adverse effects being mild and transient. Importantly, the rate of disease progression, as measured by the ALS Functional Rating Scale-Revised (ALSFRS-R) and forced vital capacity (FVC), was reduced in treated patients compared to the pretreatment period [218]. Another study involving intravenous infusion of MSCs in a transgenic ALS rat model showed delayed disease progression and preservation of motor neuron function. This was attributed to the restoration of the blood-spinal cord barrier and increased expression of neurotrophic factors such as neurturin [219]. However, a randomized placebo-controlled phase 3 study did not meet its primary endpoint of significantly slowing disease progression in ALS patients. Despite this, a pre-specified subgroup analysis suggested that MSC-NTF treatment might benefit patients with less severe disease [220].

MSCs have also shown promise in the treatment of Alzheimer’s disease (AD) through various mechanisms. Studies have demonstrated that MSCs can promote neurogenesis and synaptogenesis by secreting neurotrophic growth factors, and they can ameliorate amyloid-beta (Aβ) and tau-mediated toxicity [221]. In animal models of AD, MSCs have been shown to reduce tau phosphorylation and inflammation. For instance, intravenous administration of MSCs in the 3xTg-AD mouse model resulted in decreased tau phosphorylation and neuroinflammation, highlighting the potential of MSCs to modulate AD-like neuropathology [222]. Additionally, MSCs have been observed to reduce oxidative stress and improve synaptic plasticity, which are critical factors in the progression of AD [223]. Furthermore, MSCs can modify microglial function and suppress oxidative stress, leading to improved Aβ pathology. This is achieved through the secretion of neuroprotective and anti-inflammatory factors, as well as the transfer of functional mitochondria and miRNAs to neurons, enhancing their bioenergetic profile and promoting microglial clearance of accumulated protein aggregates [224].

In multiple sclerosis (MS), both MSCs and autologous HSCs have been explored for their ability to modulate autoimmune responses. Autologous HSC transplantation (AHSCT) has demonstrated immune “resetting” effects, leading to long-term remission in select patients with relapsing-remitting MS [225]. Clinical studies have demonstrated that AHSCT can lead to long-term remission in select patients with relapsing-remitting MS, with significant reductions in disability progression and disease activity [226,227]. MSCs have also been explored for their immunomodulatory properties in MS, but the evidence supporting their efficacy is less robust compared to AHSCT. While MSCs may offer some benefits, AHSCT remains the more established and effective option for achieving long-term remission in patients with highly active MS [228].

In Parkinson’s disease (PD), iPSC-derived dopaminergic neurons have not only provided accurate disease modeling but also shown preclinical potential for cell replacement therapy with a reduced risk of tumorigenicity compared to embryonic stem cells [5,229,230]. Specifically, studies have confirmed that iPSC-derived dopaminergic progenitors can engraft, survive, and functionally integrate into the host brain, leading to behavioral improvements in PD models [231].

NSCs are also being investigated for their potential in treating various neurological disorders, including spinal cord injury (SCI), traumatic brain injury (TBI), and stroke. NSCs are advantageous due to their lineage-restricted differentiation capacity, which reduces the risk of tumorigenicity and allows for targeted regeneration of neural tissues. Preclinical studies have shown that NSCs can promote axonal regeneration, remyelination, and protection of neuronal circuits, although clinical efficacy in humans remains to be fully established [5,232,233,234].

A summary of key preclinical and clinical studies investigating the use of various stem cell types in neurological disorders, including their therapeutic outcomes and limitations, can be found in Table 7.

Despite encouraging preclinical data, the clinical application of stem cells in neurology faces several challenges. Tumorigenicity remains a major concern, particularly with pluripotent stem cells, where even minimal uncontrolled growth in the CNS can have severe consequences, including tumor formation [243]. Immunocompatibility also limits widespread use, as allogeneic cells may trigger immune rejection, while autologous approaches are time-intensive and costly—often impractical for rapidly progressive conditions like ALS [244]. The complex and poorly regenerative environment of the CNS complicates cell survival, differentiation, and integration. Many promising results in animal models fail to translate due to species differences, disease heterogeneity, and variability in delivery routes. Additionally, ethical and regulatory constraints, especially for genetically modified or pluripotent cells, continue to slow clinical advancement.

### 3.10. Stem Cell Applications in Musculoskeletal Diseases

In musculoskeletal diseases, stem-cell based therapies offer regenerative solutions for disorders characterized by tissue degeneration, chronic inflammation, and limited self-repair capacity. Musculoskeletal conditions such as osteoarthritis (OA), rheumatoid arthritis (RA), osteoporosis, bone fractures, spinal cord injuries, and degenerative disc disease are increasingly being targeted through the application of MSCs, iPSCs, skeletal stem cells (SSCs), and ESCs. These therapies aim to restore structural integrity and function through chondrogenesis, osteogenesis, and modulation of the local immune environment.

MSCs, especially those derived from bone marrow, adipose tissue, and umbilical cord, are the most extensively studied in musculoskeletal contexts due to their differentiation capacity into bone, cartilage, tendon, and ligament lineages. BM-MSCs have been traditionally used due to their robust differentiation capacity into osteoblasts and chondrocytes, which is crucial for bone and cartilage repair [245,246]. AD-MSCs are also widely utilized, particularly for their higher clonogenicity and immunosuppressive properties, which can be beneficial in inflammatory musculoskeletal conditions [246,247]. UC-MSCs, on the other hand, exhibit superior proliferation rates and lower senescence markers, making them advantageous for long-term therapeutic applications [248].

In addition to their differentiation capabilities, MSCs secrete bioactive molecules that promote angiogenesis, reduce inflammation, and enhance matrix remodeling. For instance, UC-MSCs have been shown to secrete angiopoietin-1, which plays a significant role in reducing inflammation and promoting vascularization [248]. This paracrine activity is a key mechanism through which MSCs exert their therapeutic effects, such as in OA where studies have shown promise in improving joint function and reducing pain. A systematic review and meta-analysis by Ma et al. demonstrated significant improvements in pain and function in patients receiving MSC injections compared to control groups [249]. Another systematic review by Ha et al. reported that MSC injections led to improved clinical outcomes and evidence of cartilage repair on MRI and second-look arthroscopy in many cases [250]. The Cochrane Database of Systematic Reviews also assessed the benefits and harms of stem cell injections for knee OA. The review found that MSC injections may slightly improve pain and function up to six months post-treatment, although the evidence was of low certainty due to variability in stem cell sources and preparation methods [251]. Additionally, a study by Lee et al. showed that a single intra-articular injection of bone marrow-derived MSCs provided significant pain relief and functional improvement at nine months, with MRI evidence suggesting a preventive effect on OA progression [252]. However, it is important to note that while these studies indicate potential benefits, the evidence remains limited by small sample sizes, heterogeneity in study designs, and variability in MSC sources. Further robust clinical trials are needed to establish the long-term efficacy and safety of MSC therapies for OA.

MSCs have also shown significant promise in bone regeneration, particularly in the treatment of non-union fractures, critical-sized bone defects, and osteonecrosis. When combined with scaffolds such as hydroxyapatite or tricalcium phosphate, MSCs enhance osteogenic activity and improve biomechanical outcomes [249]. A landmark study demonstrated that autologous BM-MSCs seeded onto ceramic scaffolds led to significant cortical bone regeneration in long-bone defects [253]. Furthermore, MSCs have been successfully utilized in spinal fusion procedures and vertebral fracture repair. These applications have shown accelerated healing and reduced complications compared to conventional grafts [254,255].

Furthermore, MSCs have shown potential in enhancing tendon and ligament repair by improving collagen organization, reducing fibrosis, and promoting tissue regeneration. Preclinical studies in rotator cuff and Achilles tendon models indicate that MSCs reduce inflammation, stimulate neovascularization, and support the transformation of type III to type I collagen fibers [256]. Hooper et al. reported that bone marrow- and adipose-derived MSCs improve tissue remodeling and tendon strength, with early-phase clinical trials suggesting reduced re-tear rates, though larger randomized studies have shown mixed outcomes [257]. In Achilles tendinopathy, a phase IIa trial by Goldberg et al. demonstrated the safety and potential efficacy of autologous MSC injections, showing improvements in pain, function, and tendon morphology [258].

iPSCs offer a promising alternative for musculoskeletal tissue engineering due to their ability to differentiate into various cell types, including osteoblasts, chondrocytes, and tenocytes. For instance, iPSCs can be directed to differentiate into chondroprogenitors and subsequently into articular chondrocytes or hypertrophic chondrocytes, which can transition to osteoblasts, mimicking in vivo endochondral bone formation [259]. Additionally, iPSCs have been shown to differentiate into osteoblasts that can deposit and mineralize a collagen I extracellular matrix, further supporting their potential in bone tissue engineering [259]. Though concerns regarding tumorigenicity and genetic stability remain barriers to clinical translation.

SSCs have garnered significant attention due to their lineage-restricted differentiation capabilities and robust regenerative potential for bone and cartilage. These cells are primarily found in the bone marrow and periosteum, and recent studies have elucidated their roles in bone development, maintenance, and repair [260,261,262]. SSCs are characterized by their ability to differentiate into osteoblasts and chondrocytes, which are essential for skeletal reconstruction and the treatment of degenerative joint diseases. The therapeutic application of SSCs is still largely experimental, but promising results have been observed in preclinical models [263]. Further research is needed to fully understand the regulatory mechanisms and environmental cues that govern SSC behavior, which could lead to novel therapeutic strategies for skeletal reconstruction and degenerative joint diseases.

ESCs, though highly potent, are limited by ethical concerns and immunogenicity issues. Nevertheless, ESC-derived chondrocyte-like cells have demonstrated cartilage formation in animal models, paving the way for future therapeutic exploration [264]. Recent advancements also include the combination of stem cells with novel biomaterials to enhance therapeutic outcomes [265].

### 3.11. Stem Cell Applications in Hematology

Hematology is one of the most mature fields for stem cell applications, with HSCT established as a cornerstone therapy for a wide range of malignant and non-malignant blood disorders [266]. Stem cell therapies in hematology encompass autologous and allogeneic HSCs, as well as emerging approaches using iPSCs and gene-edited stem cells [266]. These interventions aim to reconstitute hematopoiesis, correct genetic defects, and modulate immune responses.

HSCT remains the standard of care, although not universally, for hematologic malignancies such as acute myeloid leukemia (AML), acute lymphoblastic leukemia (ALL), and multiple myeloma [267]. Allogeneic HSCT is particularly valuable for its graft-versus-leukemia (GVL) effect, which provides a potent immunologic attack against residual malignant cells [268]. This modality is often employed in AML and ALL, especially in patients with intermediate or high-risk disease profiles, as it offers curative potential by replacing diseased marrow and leveraging the GVL effect. Advances in donor registries, including the use of unrelated donors, umbilical cord blood, and haploidentical donors, have significantly expanded the pool of eligible patients [269,270,271]. Autologous HSCT is frequently utilized in relapsed lymphomas and multiple myeloma. This approach involves harvesting the patient’s stem cells before high-dose chemotherapy, which is then followed by reinfusion of the harvested cells to rescue the bone marrow. This method is associated with lower transplant-related mortality compared to allogeneic HSCT but lacks the GVL effect, which can result in higher relapse rates [272,273]. Recent advancements in conditioning regimens, such as reduced-intensity conditioning (RIC), have broadened the eligibility for HSCT by making the procedure safer for older patients and those with comorbidities [269,270]. Additionally, improvements in post-transplant immunosuppression and supportive care have contributed to better survival outcomes and reduced transplant-related mortality [271,274].

HSCT is also widely used in non-malignant hematologic disorders, including aplastic anemia, myelodysplastic syndromes (MDS), sickle cell disease (SCD), and β-thalassemia [275]. For inherited hemoglobinopathies like SCD and β-thalassemia, allogeneic HSCT, particularly from matched sibling or haploidentical donors, has shown significant promise [276]. Clinical trials and studies have demonstrated that HSCT can lead to transfusion independence and reversal of disease-related organ damage in these populations. For instance, in patients with severe SCD, nonmyeloablative HLA-matched sibling allogeneic HSCT has shown high rates of stable donor engraftment and significant improvements in organ function, with a reduction in hospitalization rates and narcotic use [277]. Similarly, for β-thalassemia, HSCT has achieved disease-free survival rates exceeding 90% in children with favorable risk profiles and matched sibling donors [278]. The use of haploidentical donors has expanded the donor pool significantly, making curative therapy more accessible. Recent advancements in haploidentical HSCT protocols have shown promising outcomes with high overall survival rates and reduced incidences of graft-versus-host disease (GVHD) [279,280]. These protocols include nonmyeloablative conditioning regimens, which are particularly beneficial for adults with preexisting organ damage, allowing for safer transplantation with reduced toxicity [281].

iPSC technology is indeed being explored in hematology for various applications, including disease modeling, drug screening, and the potential generation of patient-specific HSCs for transplantation. iPSCs have been utilized to study disease mechanisms in conditions such as Fanconi anemia, Diamond-Blackfan anemia, and myeloproliferative neoplasms [282,283,284,285]. Despite significant progress, the differentiation of iPSCs into functional long-term repopulating HSCs remains a challenge. Current research focuses on optimizing differentiation protocols and understanding the molecular mechanisms underlying hematopoiesis to improve the efficiency and functionality of iPSC-derived HSCs [286,287,288]. For instance, recent studies have shown that specific culture conditions and the use of transcription factors can enhance the generation of engraftable HSCs from iPSCs [286]. iPSCs offer a valuable platform for personalized hematology by enabling the creation of patient-specific cell lines that can be used to model hematologic diseases and screen potential therapeutic agents. This approach allows for a better understanding of disease pathogenesis and the identification of novel drug targets [282,283,289]. Additionally, iPSCs combined with gene-editing technologies, such as CRISPR-Cas9, hold promise for correcting genetic defects in patient-derived cells, potentially leading to autologous cell-based therapies [283,290].

These gene-edited stem cell therapies are also explored in monogenic blood cell disorders such as SCD and β-thalassemia. Exagamglogene autotemcel (exa-cel, formerly CTX001) is a CRISPR-based therapy that targets the BCL11A gene to reactivate fetal hemoglobin (HbF) production. This approach has demonstrated significant clinical benefits. In a phase 3 study, exa-cel enabled transfusion independence in 91% of patients with transfusion-dependent β-thalassemia, with a mean total hemoglobin level of 13.1 g/dL and a mean fetal hemoglobin level of 11.9 g/dL during transfusion independence [291]. Similarly, in patients with SCD, exa-cel eliminated vaso-occlusive crises in 97% of patients for at least 12 consecutive months [292]. The mechanism involves CRISPR-Cas9-mediated editing of the erythroid-specific enhancer region of BCL11A in autologous CD34+ HSCs, which leads to the reactivation of γ-globin synthesis and increased HbF levels. This strategy leverages the curative potential of hematopoietic stem cell transplantation while avoiding the immunologic complications associated with allogeneic transplantation [293].

Ongoing advances aim to expand the accessibility, safety, and efficacy of stem cell therapies in hematology. Development of universal donor iPSC lines and gene-edited immune-evasive cells may eliminate the need for HLA matching. Innovations in non-viral gene delivery, such as base and prime editing, are improving the precision of genome engineering. Additionally, ex vivo HSC expansion using small molecules or niche-mimicking conditions may improve engraftment and reduce time to hematologic recovery.

A summary of representative stem cell therapies utilized across malignant and non-malignant hematologic disorders, along with associated outcomes, is presented in Table 8.

### 3.12. Oncology

Cancer remains one of the leading causes of mortality worldwide. Despite advances in chemotherapy, radiotherapy, and immunotherapy, challenges like systemic toxicity, tumor resistance, and limited drug penetration hinder treatment efficacy. Stem cell-based therapies can address these limitations through their tumor-homing capabilities and ability to deliver therapeutic agents directly to tumor sites [301]. This section highlights key mechanisms under investigation: apoptosis induction, tumor microenvironment modulation, exosome-based delivery, and integration with conventional therapies.

#### 3.12.1. Induction of Tumor Cell Apoptosis

A major strategy involves engineering MSCs and NSCs to express pro-apoptotic ligands such as Tumor Necrosis Factor-Related Apoptosis-Inducing Ligand (TRAIL) or FasL, which selectively trigger cell death in malignant cells. MSCs, adipose- and iPSC-derived MSCs expressing TRAIL showed tumor suppression across models of squamous cell, cervical, lung, and glioma cancers [302,303,304,305,306,307]. Therapeutic efficacy was enhanced when combined with chemotherapy or small molecule inhibitors targeting TRAIL-resistant cancer stem cells [308,309]. FasL-expressing MSCs also showed tumor suppression in multiple myeloma models [310]. A Phase I/II clinical trial is underway assessing TRAIL-expressing MSCs with standard care in metastatic lung adenocarcinoma [311,312]. Genetically modified stem cells have also been applied in suicide gene therapy, converting prodrugs into cytotoxic agents at tumor sites [313].

#### 3.12.2. Stem Cell-Derived Exosomes as Anti-Tumor Agents

Modified stem cells can release exosomes—extracellular vesicles carrying bioactive molecules—that shape the tumor microenvironment [314]. MSC-derived exosomes loaded with microRNAs have enhanced chemosensitivity and immune response. In hepatocellular carcinoma, exosomes carrying miR-374c-5p suppressed epithelial–mesenchymal transition [315]. Similarly, in esophageal and ovarian cancers, miR-655-3p and miR-424 suppressed tumorigenesis and angiogenesis [316,317]. Compared to direct miRNA infusion, this method offers improved stability and reduced off-target effects.

#### 3.12.3. Immunomodulation of Tumor Microenvironment

Solid tumors often evade immunity through immunosuppressive microenvironments. Genetically engineered stem cells can reverse this by delivering immune-activating cytokines. For example, MSCs expressing interferon-alpha increased T cell infiltration and synergized with checkpoint inhibitors [318]. A phase I clinical trial in ovarian cancer, intraperitoneally delivered IFN-β-secreting MSCs activated immune responses without infusion-related toxicity [319]. Similarly, MSCs carrying CXCL9 and OX40L promoted immune cell infiltration in colon cancer, while IL-7/IL-12-expressing MSCs enhanced CAR T cell responses in solid tumor models [320,321]. This stem cell-based delivery can overcome poor immune infiltration seen with direct therapies and signals a shift toward cell-based immunotherapy.

#### 3.12.4. Stem Cells as Drug Delivery Platforms

Due to their tumor-tropic migration and efflux protein expression, MSCs are also being explored as chemotherapeutic delivery vehicles [322,323,324]. MSCs preloaded with nanoparticle-conjugated drugs such as paclitaxel showed greater tumor accumulation than direct intravenous administration [322]. Further advancements in click-chemistry have improved targeted payload capacity, though concerns remain around biosafety and nanoparticle stability [325,326,327].

#### 3.12.5. Tumor-Specific Applications and Translational Highlights

Stem cell mechanisms show promise in hard-to-treat tumors like gliomas, where NSCs can bypass the blood–brain barrier (BBB) [328]. A phase I trial confirmed the safety of stem cell-mediated oncolytic therapy in glioblastoma, though clinical efficacy was modest [329]. In another trial, suicide gene-expressing MSCs demonstrated localized prodrug conversion with minimal toxicity in glioblastoma patients [330].

In pancreatic cancer models, MSCs expressing IFN-β and herpes simplex virus thymidine kinase coadministered with ganciclovir reduced tumor growth and metastasis [331,332]. One study found gemcitabine-exposed MSCs could induce chemoresistance via CXCR3 signaling, but nanovesicles carrying antagonists successfully reversed this effect [333].

Stem cell-based therapies represent a dynamic platform for cancer treatment. Leveraging tumor-homing properties and genetic malleability, stem cells can deliver therapies with spatial accuracy, induce apoptosis, modulate immunity, and transport drugs. While preclinical models show promise across tumor types, clinical translation faces challenges including tumorigenicity risk, therapeutic durability, and scalability. As trials expand and delivery methods improve, stem cells are likely to play an increasing role in next generation therapeutics.

### 3.13. Gastrointestinal Diseases

Gastrointestinal (GI) diseases pose a substantial clinical burden worldwide. Stem cell therapy has shown considerable promise in the treatment of various GI disorders, including inflammatory bowel disease, liver cirrhosis, and gastric ulcers, owing to their regenerative capabilities and immunomodulatory properties. This section explores recent clinical and preclinical advances in stem cell applications across key gastrointestinal conditions.

#### 3.13.1. Inflammatory Bowel Disease (IBD)

Perianal fistulas in Crohn’s disease remain one of the most treatment-resistant manifestations, often unresponsive to antibiotic therapies and requiring surgery [334]. MSCs derived from bone marrow or adipose tissue, offer a novel therapeutic alternative due to their ability to modulate immune responses and promote tissue repair. Clinical trials have demonstrated efficacy. One phase I trial using autologous MSCs led to complete healing in 83% of patients [335]. Similar success was observed with adipose-derived MSCs, showing a 75% complete healing rate [336]. A comparative study demonstrated superior fistula closure and reduced relapse rates in MSC-treated patients [337]. Pediatric trials mirrored these outcomes, achieving 83% clinical and radiographic healing [338].

Across studies, adverse events were minimal. The therapeutic effects are attributed to MSC-mediated suppression of proinflammatory cytokines and enhancement of regulatory T cells and IL-10 production, fostering tissue regeneration [339]. These findings support MSC therapy as a promising adjunct in Crohn’s disease management.

Stem cell-based therapies have also shown growing promise in the treatment of ulcerative colitis. In a phase I/II clinical trial, umbilical cord-derived MSC infusion, given alongside standard treatment, improved clinical symptoms, mucosal healing, histologic inflammation, and quality of life in patients with moderate to severe ulcerative colitis [340]. Similar results were observed in a recent multicenter, double-blind, placebo-controlled phase II trial, where umbilical cord-derived MSC infusion led to higher rates of clinical remission and mucosal healing in 41 patients with moderate-to-severe ulcerative colitis unresponsive to conventional therapies, including glucocorticoids; 34.2% achieved clinical remission at 6 months, with no serious treatment-related adverse events reported [340]. A 2019 meta-analysis of eight preclinical and seven clinical studies found that MSC therapy improved healing rates, reduced disease activity, and demonstrated an excellent safety profile in the treatment of ulcerative colitis [341]. These findings support the therapeutic potential and safety of MSC-based interventions in ulcerative colitis, underscoring the need for larger, standardized clinical trials.

#### 3.13.2. Liver Disease (Cirrhosis, ALF, NASH)

Liver cirrhosis and acute liver failure (ALF) are critical conditions in which the liver’s native regenerative capacity falls short, often leaving transplantation as the only curative option [342]. In this context, MSCs—particularly from bone marrow and umbilical cord—have shown capacity to restore liver function. Multiple early-phase trials report encouraging outcomes. One phase I–II study in patients with hepatitis or alcoholic cirrhosis showed significant reductions in Model for End-Stage Liver Disease scores post-MSC injection [343]. Other trials noted reduced bilirubin and transaminase levels, and increased albumin and prothrombin, alongside improved quality of life [343,344]. In murine models of non-alcoholic steatohepatitis, MSC treatment decreased hepatic inflammation and triglyceride accumulation [345]. A meta-analysis of 12 randomized controlled trials confirmed sustained improvements in liver function, especially in hepatitis B (HPV)-associated disease [346]. Notably, one study reported increased long-term survival in HBV-related cirrhosis [347]. The therapeutic effects arise from a combination of paracrine signaling, immune modulation, homing to damaged tissue, and differentiation into hepatocyte-like cells [348]. Together, these findings support the continued investigation of MSCs as a feasible therapeutic strategy for hepatic diseases.

#### 3.13.3. Short Bowel Syndrome/Intestinal Failure

Short bowel syndrome (SBS) and intestinal failure present major clinical challenges, particularly in pediatric populations. Conventional management is limited to supportive care and intestinal transplantation, both with significant complications [349]. While stem cell-based approaches offer regenerative potential, much of the research remains preclinical. In a foundational preclinical study on SBS, intestinal scaffolds seeded with embryonic or induced pluripotent stem cells (iPSCs) promoted partial intestinal regeneration, though the resulting tissue lacked enteric nervous system integration [350]. Additional studies using MSCs in radiation-induced intestinal injury models have shown improvements in crypt regeneration, epithelial repair, and survival [351,352]. A meta-analysis involving ischemia–reperfusion injury reported reductions in inflammation and oxidative stress, along with improved survival [353]. Despite these advances, translation remains limited by hurdles such as engraftment failure and incomplete functional maturation [354]. Addressing these challenges will be key to advancing stem cell applications in SBS.

#### 3.13.4. Gastric Disorders

Stem cell-based therapy has also shown efficacy in the regenerative treatment of gastric ulcers and gastroparesis. Several preclinical studies have demonstrated that MSC transplantation can accelerate both gastric and peptic ulcer healing, promote angiogenesis and re-epithelization, and reduce inflammatory infiltrates [355,356,357,358]. These effects are largely attributed to paracrine secretion of growth factors, like vascular endothelial growth, and anti-inflammatory cytokines [359].

In gastroparesis models, neural stem cell transplantation restored gastric motility through the secretion of nitric oxide, an inhibitory neurotransmitter critical for gastric function [360]. While these studies demonstrate therapeutic promise, further work is needed to validate efficacy, assess long-term safety, and optimize delivery strategies in human models.

There is growing evidence that stem cell-based therapies offer therapeutic potential in the treatment of various gastrointestinal conditions, with the most robust clinical success seen in perianal Crohn’s disease. Applications in liver cirrhosis show promise, while interventions for short bowel syndrome and gastric disorders remain largely preclinical. Future directions include enhancing engraftment strategies, incorporating stem cell-derived organoids, and optimizing delivery platforms to bridge the gap between experimental models and clinical translation.

## 4. Ethical and Regulatory Considerations

Trials have demonstrated the therapeutic potential for stem cell-based therapy, but their clinical application raises significant ethical and regulatory concerns that remain unresolved. The use of hESCs, in particular, poses challenges since harvesting hESCs requires the destruction of embryos. This raises significant ethical concern for many. Beyond these ethical debates, safety risk also limits the use of hESC-based therapies. Because hESCs possess high plasticity and the capacity to differentiate into multiple cell types, they carry an increased risk of uncontrolled growth in transplantation settings [361].

iPSCs are considered more ethically acceptable since their derivation does not involve destruction of embryos, however they are not without risk. Similarly to hESCs, iPSCs exhibit genomic instability and incomplete differentiation which can limit their safety in transplantation [361].

MSCs are generally viewed as less ethically controversial than embryonic stem cells since their generation does not require embryo destruction. Nonetheless, ethical concerns persist regarding informed consent and ownership of perinatal tissues such as umbilical cord blood and placental samples. Additionally, across all stem cell types, there are significant concerns surrounding equitable access, as novel therapies risk widening existing healthcare disparities [362].

Globally, stem cell-based therapies are regulated differently across countries to ensure patient safety and research integrity. In the United States, stem cell-based therapies are regulated at both the federal and state levels primarily by the Food and Drug Administration (FDA) and the National Institutes of Health (NIH). Such regulations include rigorous preclinical validation, standardized manufacturing practice, and close monitoring of clinical trials before approval. Despite these efforts, the rise of stem cell clinics that are unregulated suggests the need for stricter oversight and transparent reporting. Ongoing collaboration between clinicians, researchers, regulatory authorities, and bioethics will be critical to balancing innovation with patient safety, ethical responsibility, and equitable access. 

## 5. Conclusions and Future Directions

In conclusion, the emerging literature surrounding stem cell-based therapies demonstrates great promise, with its applications extending across multiple systems and diseases. Among systems, cardiovascular, dermatology and ophthalmology had relatively stronger evidence, whereas neurology, orthopedics, pulmonology, nephrology and endocrinology require more larger controlled studies. Among stem cell subtypes, mesenchymal stem cells are the most frequently studied, are accessible and have a favorable safety profile. Induced pluripotent stem cells are also promising for future precision medicine, though safety and scalability challenges remain apparent.

Limitations of this review include significant heterogeneity in stem-cell sources, preparation methods, and routes of administration. These complications make comparison across studies difficult. Furthermore, long-term safety data remains uncertain; standardized manufacturing protocols and regulatory agencies are required to ensure quality, safety, and comparability across trials.

Future research should prioritize standardizing stem-cell preparation and reporting processes, as well as large-scale prospective trials to validate efficacy and scalability of outcomes. Many studies in this review are small, early-stage studies with high heterogeneity, making it difficult to draw concrete conclusions. Furthermore, future studies should address the cost and scalability of stem cell therapies to ensure appropriate and equitable clinical adoption. By addressing these gaps, clinical uptake and utility will be improved in the future, ensuring efficacious and safe regenerative therapies across fields.

## Figures and Tables

**Figure 1 ijms-26-09659-f001:**
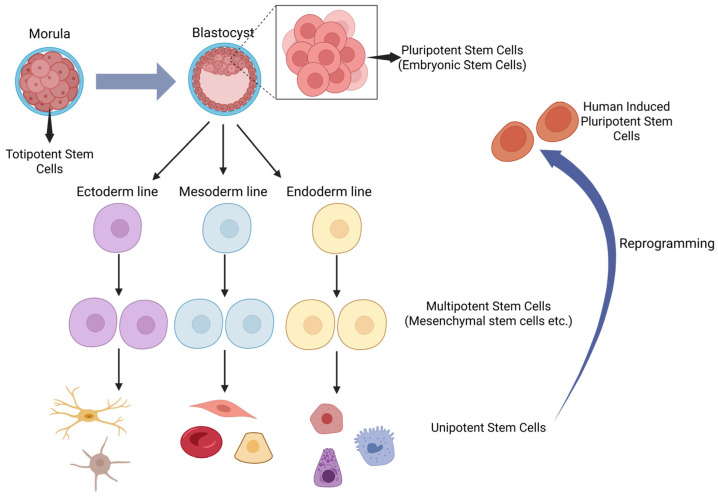
Differentiation Stages of Stem Cells.

**Table 1 ijms-26-09659-t001:** Comparison of stem cell types. Abbreviations: OCT4: Octamer Transcription Factor 4, SOX2: SRY-Box Transcription Factor 2.

Stem Cell Type	Differentiation Potential	Source(s)	Key Markers	Clinical Relevance/Applications	Limitations/Risks	References
Embryonic Stem Cells (ESCs)	Pluripotent (all somatic cell types)	Inner cell mass of blastocyst	OCT4, SOX2, NANOG	Regenerative medicine, disease modeling	Ethical concerns, teratoma risk, immunogenicity	[5,6,7]
Induced Pluripotent Stem Cells (iPSCs)	Pluripotent (all somatic cell types)	Reprogrammed adult somatic cells (e.g., skin, blood, urine)	OCT4, SOX2, NANOG	Disease modeling, drug screening, potential for autologous cell therapy	Tumorigenicity, genetic/epigenetic instability	[5,6,8]
Hematopoietic Stem Cells (HSCs)	Multipotent (all blood cell lineages)	Bone marrow, peripheral blood, umbilical cord blood	CD34+, CD133+	Hematopoietic reconstitution (e.g., transplantation for leukemia, anemia)	Limited to blood lineages, graft-vs-host disease	[9,10,11]
Mesenchymal Stem Cells (MSCs)	Multipotent (bone, cartilage, fat, muscle)	Bone marrow, adipose tissue, umbilical cord, placenta, urine	CD73+, CD90+, CD105+	Tissue repair, immunomodulation, autoimmune and degenerative disease therapy	Heterogeneity, variable efficacy, rare tumorigenicity	[10,12,13,14]

**Table 2 ijms-26-09659-t002:** Sample clinical studies on the use of stem cells in cardiovascular diseases, categorized by stem cell type. Abbreviations: CDC: Cardiosphere-Derived Cells, LV: Left Ventricle, LVEF: Left Ventricular Ejection Fraction, MACE: Major Adverse Cardiovascular Events, BM-MSCs: Bone-Marrow Mesenchymal Stem Cells, LVESV: Left ventricular end-systolic volume, UC-MSCs: Umbilical Cord Mesenchymal Stem Cells, ADSCs: Adipose Tissue Derived Stem Cells, HSCs: Hematopoietic Stem Cells, ECG: Electrocardiogram, EPCs: Endothelial Progenitor Cells, SC: Stem Cell, hESCs: Human Embryonic Stem Cells, iPSCs: Induced Pluripotent Stem Cells.

Stem Cell Type	Study	Outcomes
Allogeneic Cardiac Stem Cells (Progenitor Cells)	Makkar et al. [21]	There was no difference in wound size proportionate to baseline between the CDC and placebo groups. However, significant reductions in LV end-diastolic volume, LV end-systolic volume, and N-terminal pro b-type natriuretic peptide were observed in patients treated with CDC compared to placebo. There were no deaths related to treatment. The authors also noted that allogeneic therapy has an advantage over autologous therapy as it does not require myocardial biopsy and is readily available.
Autologus Cardiac Stem Cells	Makkar et al. [23]	Patients treated with CDCs showed reduced scar tissue, increased viable heart mass, and improved regional contractility and regional systolic wall thickening compared to controls. However, changes in end-diastolic volume, end-systolic volume, and LVEF during follow-up did not differ between the groups. No patients died during follow-up, nor did they develop cardiac tumors or MACE in either group.
Skeletal Myoblasts	Miyagawa et al. [54]	The majority of patients with ischemic cardiomyopathy showed significant symptomatic improvement. There was also a significant decrease in serum brain natriuretic peptide, pulmonary artery pressure, pulmonary capillary wedge pressure, pulmonary vein resistance, and left ventricular wall stress. However, the efficacy was not as good as that in patients with dilated cardiomyopathy. No major procedure-related complications were observed during follow-up.
	Menasche et al. [33]	Myoblast transfer did not enhance regional or global LV function beyond what was observed in control patients. However, patients receiving the high-dose cell line showed a significant reduction in LV volumes compared to placebo. Myoblast-treated patients experienced a higher number of arrhythmias.
BM-MSCs	Mathiasen et al. [55]	LVESV was significantly reduced in the MSC group compared to placebo. Significant improvements were also observed in LVEF, stroke volume, and myocardial mass. Additionally, the MSC group experienced a reduction in scar tissue and an improvement in quality of life scores, unlike the placebo group. Angina hospitalizations were significantly fewer in the MSC group at long-term follow-up. No adverse events were identified.
UC-MSCs	Bartolucci et al. [56]	The UC-MSC-treated group demonstrated significant improvements in left ventricular ejection fraction and quality of life. There were no differences compared to the control group in terms of mortality, heart failure hospitalizations, arrhythmias, or new malignancies.
ADSCs	Qayyum et al. [57]	Intramyocardial ADSC treatment was safe but did not improve exercise capacity compared to placebo. However, exercise capacity increased after treatment with ADSC compared to baseline but not in the placebo group.
HSCs	Aceves et al. [58]	Improvements were observed in left ventricular ejection fraction, stress ratio values, stress tests, and the number of affected segments of the left ventricle. After 6 months, ECG results were normal in all patients.
EPCs	Steinhoff et al. [52]	Although a reduction in scar size and nonviable tissue, along with an improvement in segmental myocardial perfusion, were observed, no significant difference in LVEF was detected after CD133+ SC injection compared to placebo.
hESCs	Menasche et al. [34]	All patients experienced symptomatic improvement, and systolic motion increased in treated segments. No serious treatment-related side effects occurred. hESCs can differentiate into cardiovascular progenitors after transplantation in patients with severe ischemic LV dysfunction.
iPSCs	Kawamura et al. [40]	Improvements were observed in heart failure symptoms, left ventricular contractility, and myocardial blood flow. No treatment-related adverse events occurred.

**Table 3 ijms-26-09659-t003:** Key clinical studies on the use of stem cells in dermatological diseases. Abbreviations: HSCT: Hematopoietic Stem Cell Therapy, UC-MSCs: Umbilical Cord Mesenchymal Cells, PASI: Psoriasis Area and Severity Index, ADMSCs: Adipose-Derived Mesenchymal Stem Cells, UCB-MSCs: Umbilical Cord Blood Derived Mesenchymal Stem Cells, EASI: Eczema Area and Severity Index, BM-MSCs: Bone Marrow Derived Mesenchymal Stem Cells, IL-17: Interleukin-17, AD: Atopic Dermatitis, SCORAD: SCORing Atopic Dermatitis, ADSCs: Adipose Derived Stem Cells, BM-HSC: Bone Marrow-Derived Hematopoietic Stem Cells, C7: Complement 7, BEBSS: Birmingham Epidermolysis Bullosa Severity Score, DMSCs: Dermal Mesenchymal Stem Cells, AA: Alopecia Areata, AGA: Androgenetic Alopecia, HFSCs: Hair Follicle Stem Cells, AFSCs: Adipose Tissue Derived Follicle Stem Cells, SVF: Stromal Vascular Fraction, SSc: Systemic Sclerosis.

Disease/Condition		Stem Cell Therapy Utilized	Study	Outcomes
Psoriasis		HSCT	Kaffenberger et al. [63]	Allogeneic transplantation has a longer remission period than autologous transplantation but has higher mortality rates.
		UC-MSCs	Cheng et al. [66]	In 47.1% of patients, there was at least a 40% improvement in the PASI score. Allogeneic UC-MSC treatment is safe and partially effective for patients with psoriasis, and Treg levels may serve as a biomarker to predict treatment efficacy.
		ADMSCs	Bajouri et al. [69]	No significant side effects were observed during the 6-month follow-up, and the PASI score and appearance of lesions were reported to have decreased in most patients.
Atopic Dermatitis		UCB-MSCs	Kim et al. [74]	In the high-dose hUCB-MSC-treated group, 55% of patients showed a 50% decrease in EASI score. Serum IgE levels, blood eosinophil counts, and pruritus scores decreased significantly. No serious adverse events reported.
		BM-MSCs	Shin et al. [76]	Eighty percent of patients achieved EASI-50, with no serious side effects observed. MSC therapy may be a promising option for patients with moderate to severe atopic dermatitis, particularly those with elevated IL-17 levels.
			Seo et al. [77]	A significantly higher EASI-50 rate was achieved compared to the control group, with no serious side effects. However, there were no significant differences in pruritus scores and quality of life index.
		ADMSCs	Ra et al. [72]	Partial decrease in SCORAD scores of 4 patients and no serious side effects were observed.
Vitiligo		Cell Transplantation with Hair Follicle Origin	Mohanty et al. [78]	Nine of 14 patients achieved >75% repigmentation. The transplantation procedure is recommended for patients with stable vitiligo for at least 1 year.
		Combined Cell Transplantation with Epidermal and Hair Follicle Origin	Ramos et al. [96]	Of the 24 patients, 25% showed an excellent repigmentation rate and 50% showed a good repigmentation rate. The best results were in the face and neck lesions, with better responses observed in patients with segmental vitiligo than in those without segmental vitiligo.
		ADSCs with Melanocyte Culture	Saleh et al. [97]	The study indicated that culturing stem cells derived from adipose tissue alongside melanocytes from hair follicles could be a safe and effective treatment for patients with stable localized vitiligo resistant to other therapies.
Epidermolysis Bullosa		BM-HSC	Wagner et al. [98]	Five of six recipients showed increased C7 deposition at the dermal-epidermal junction, although there was no normalization of connective fibers. One recipient died from graft rejection and infection. None of the patients had detectable anti-C7 antibodies. It is recommended that treatment be initiated with a cost–benefit analysis.
		BM-MSCs	Petrof et al. [99]	A regression in the Birmingham Epidermolysis BEBSS and general severity score was observed in all patients. No side effects necessitating treatment discontinuation were reported.
		ABCB5+ DMSCs	Kiritsi et al. [100]	A 13.0% decrease in the Epidermolysis Bullosa Disease Activity and Scarring Index score and an 18.2% decrease in the Epidermolysis Bullosa Clinician-Researcher Results Scoring Tool were observed. Significant reductions in pruritus and pain scores were also noted. Hypersensitivity reactions occurred in 2 out of 16 patients, resolving after treatment discontinuation.
		UCB-MSCs	Lee et al. [101]	Improvements were noted in the BEBSS, body surface area involvement, blister count, pain, pruritus reduction, and quality of life, with no serious treatment-related side effects reported.
Alopecia	AA	ADSCs	Anderi et al. [102]	All patients exhibited increased hair density and diameter, along with decreased pull-test results.
		BM-MSCs	Elmaadawi et al. [103]	40 patients (20 AA and 20 AGA) included in the study. All patients demonstrated significant improvement with no serious side effects observed.
		UCB-MSCs (Stem Cell Educator Therapy)	Li et al. [89]	Treatment resulted in significant improvement in all patients, which persisted for 2 years, with no serious side effects observed.
	AGA	Autologous HFSCs	Gentile et al. [84]	The average hair count and density increased from baseline values, with a reported 29% average increase in hair density in the treated area.
		AFSCs	Gentile et al. [85]	Patients exhibited improvements in hair density, compared to baseline values in the treated area. Scalp biopsy evaluations revealed an increase in the number of hair follicles. HD-AFSCs in micrografts may offer a safe and effective alternative treatment option for hair loss.
		ADSCs	Tak et al. [104]	A greater increase in hair count obtained in the treatment group than in the control group. A significant improvement in hair diameter was also noted in the treatment group.
			Kim et al. [105]	The increase in hair density on the SVF-treated side compared to the untreated side was statistically significant. However, while an increase in thickness was observed, it was not statistically significant.
		BM-MSCs	Elmaadawi et al. [103]	-
SSc		HSCT	Oyama et al. [106]	A statistically significant improvement in the Modified Rodnan skin score was achieved. However, cardiac and renal functions remained stable. Overall and progression-free survival rates were 90% and 70%, respectively. The study showed that non-myeloablative autologous HSCT had similar success to the myeloablative regimen, with fewer side effects and no toxicity.
		ADSC	Granel et al. [107]	The study evaluated injection of autologous SVF cells into the hands of 12 patients with systemic sclerosis and resulted in significant improvements in hand disability and pain, Raynaud phenomenon, finger edema, and quality of life. No serious adverse events occurred during the procedure or follow-up.

**Table 4 ijms-26-09659-t004:** Key clinical studies on the use of stem cells in ophthalmic diseases. Abbreviations: CLET: Cultivated Limbal Epithelial Transplantation, ADR: Adverse Drug Reactions, CALEC: Cultivated Autologous Limbal Epithelial Cell, LSCD: Limbal Stem Cell Deficiency, BM-MSCs: Bone Marrow Mesenchymal Cells, iPSC: Induced Pluripotent Stem Cells, LSC: Limbal Stem Cell, hESCs: Human Embryonic Stem Cells, RPC: Retinal Progenitor Cell, RPE: Retinal Pigment Epithelium.

Study	Stem Cell Type	Ophthalmic Disorder	Outcome
Fasolo et al. (2017) [165]	Cultured Limbal stem cells	Limbal Stem Cell Deficiency	One year after surgery, 41% of the 59 primary CLET procedures were successful, 39% partially successful and 20% failed. The most common ADRs recorded for the primary unsuccessful CLETs were ulcerative keratitis, melting/perforation, and epithelial defects/disepithelialisation.
Sangwan et al. (2011) [166]	Limbal epithelial cells	Limbal Stem Cell Deficiency	A completely epithelised, avascular and clinically stable corneal surface was seen in 142 of 200 (71%) eyes at a mean follow-up of 3 ± 1.6 (range: 1–7.6) years; improvement in visual acuity in 60.5% of eyes.
Rama et al. (2010) [167]	Limbal epithelial cells	Limbal Stem Cell Deficiency	Permanent restoration of a transparent, renewing corneal epithelium was attained in 76.6% of eyes.
Jurkunas et al. (2025) [169]	Limbal epithelial cells	Limbal Stem Cell Deficiency	Transplantation of CALEC constructs in patients with both mild and severe forms of LSCD achieved corneal surface restoration with limbal epithelial cells and improved clinical symptoms
Miotti et al. (2021) [168]	Limbal tissue autograft transplantation (LSCs), iPSCs, MSCs, BM-MSCs	Limbal Stem Cell Deficiency	Improved visual acuity, rapid surface healing, stable epithelial adhesion without recurrent erosion, arrest or regression of corneal neovascularization
LIu et al. (2024) [185]	hESC, iPSCs	Age related macular degeneration	Slower rates of disease progression
Sharma et al. (2019) [175]	iPSCs	Age related macular degeneration	RPE patches have been derived from patient-specific iPSCs and applied on a scaffold to support long-term cell viability and function
Chen et al. (2023) [186]	RPCs, MSCs, hESCs-RPE, iPSCs-RPE	Retinitis pigmentosa	Improvement in best corrected visual acuity
Chen et al. (2023) [186]	RPCs, MSCs, hESCs-RPE, iPSCs-RPE	Stargardt disease	Improvement in best corrected visual acuity
Moghadam Fard et al. (2023) [182]	hESCs, BM-MSCs, MSCs	Stargardt disease	Improvement in visual acuity

**Table 5 ijms-26-09659-t005:** Key clinical studies on the use of stem cells in Pulmonary diseases. Abbreviations: AD-MSCs: Adipose Derived Mesenchymal Cells, BM-MSCs: Bone Marrow Mesenchymal Cells, BSC: Bronchial Stem Cell, CF: Cystic Fibrosis, COPD: Chronic Obstructive Pulmonary Disease, UC-MSCs: Umbilical Cord Mesenchymal Cells.

Study	Stem Cell Type	Pulmonary Disorder	Outcome
Le Thi Bich et al. (2020) [188]	UC-MSCs	Chronic Obstructive Pulmonary Disease	Therapy was well-tolerated with no infusion-related toxicities or serious adverse effects, and treated patients had significantly fewer acute exacerbations over 6 months.
Rio et al. (2023) [189]	AD-MSCs, UC-MSCs	Chronic Obstructive Pulmonary Disease	COPD AD-MSC were as efficient in reducing elastase-induced lung emphysema, UC-MSC reduced lung emphysema, equal therapeutic potential of AD-MSC from COPD and non-COPD subjects in the pre-clinical model
Wu et al. (2022) [187]	Autologous bronchial BSCs,	Chronic Obstructive Pulmonary Disease	Alleviation of symptoms and pulmonary function enhancement in patients receiving transplantation, procedure was safe and well-tolerated
Campo et al. (2021) [191]	BM-MSCs	Idiopathic Pulmonary Fibrosis	The treatment was well tolerated, though no significant improvements in pulmonary function were observed
Liu et al. (2025) [192]	Basal Cells	Idiopathic Pulmonary Fibrosis	Preclinical study utilizing autologous proximal airway basal cells demonstrated fibrotic repair and alveolar regeneration in bleomycin-injured mice
Matthay et al. (2018) [194]	BM-MSCs	Acute Respiratory Distress Syndrome	No improvement in outcomes, but the infusion was safe and well tolerated
Regmi et al. (2024) [195]	UC-MSCs, BM-MSCs, and AD-MSCs	Acute Respiratory Distress Syndrome	Umbilical cord MSCs showed superior efficacy to MSCs from bone marrow or adipose tissue. These mice had higher survival, improved lung histology, and less alveolar damage
Bonfield and Lazarus (2025) [196]	MSCs	Cystic Fibrosis	Allogeneic MSCs were safe and play a role in reducing infections and inflammation in CF patients
Roesch et al. (2023) [198]	BM-MSCs	Cystic Fibrosis	No dose-limiting toxicities, deaths or life-threatening adverse events were observed, no significant functional change observed
Sutton et al. (2017) [197]	MSCs	Cystic Fibrosis	MSCs secrete supernatants that are anti-inflammatory and anti-microbial and have potential in CF

**Table 6 ijms-26-09659-t006:** Key clinical studies on the use of stem cells in Nephrology diseases. Abbreviations: AD-MSCs: Adipose Derived Mesenchymal Cells, BM-MSCs: Bone Marrow Mesenchymal Cells, CKD: Chronic Kidney Disease, HSCT: Hematopoietic Stem Cell Therapy, IL-2: Interleukin-2, UC-MSCs: Umbilical Cord Mesenchymal Cells, SLE: Systemic Lupus Erythematosus, SLEDAI: Systemic Lupus Erythematosus Disease Activity Index, ANA: Anti-Nuclear Antibodies, anti-dsDNA: Anti-double stranded DNA.

Study	Stem Cell Type	Nephrology Disorder	Outcome
Maklough et al. (2018) [199]	BM-MSCs	Chronic Kidney Disease	Improved renal function and structure in preclinical models, safety and tolerability were established in patients with CKD
Salybekov et al. (2024) [201]	MSCs	Acute Kidney Injury and Chronic Kidney Disease	Clinical trials have shown that MSCs are safe and well-tolerated in both chronic and acute kidney disease, but efficacy results have been modest
Wang et al. (2022) [200]	MSCs	Chronic Kidney Disease	Preclinical trials have indicated that MSCs can slow progress of diabetes mellitus, decrease mesangial thickening and macrophage infiltration in mice models
Burt et al. (2006) [210]	HSCT	Refractory Systemic Lupus Erythematosus	Stabilization of renal function and significant improvements in SLEDAI score, ANA, anti-ds DNA, complement levels, and lung diffusing capacity were reported. Treatment-related mortality was 2%. Overall 5-year survival was 84%, and dis-ease-free survival 5 years post-HSCT was 50%.
Sun et al. (2009) [211]	BM-MSCs	Refractory Systemic Lupus Erythematosus	Four SLE patients resistant to immunosuppressive therapy were included in the study, and all treated patients achieved remission lasting 12–18 months. Improvements in disease activity, serological markers, and renal function were observed.
Deng et al. (2017) [205]	UC-MSCs	Refractory Systemic Lupus Erythematosus	hUC-MSC has no apparent additional effect over and above standard immunosuppression.
Kamen et al. (2022) [207]	UC-MSC	Refractory Systemic Lupus Erythematosus	UC-MSC infusions were safe and may have efficacy in lupus
Liang et al. (2025) [203]	MSCs	Refractory Systemic Lupus Erythematosus	Patients clinically improved with decreased in SLEDAI score and 24 h proteinuria, Anti-dsDNA levels decreased, improvement in glomerular filtration rate
Ranjbar et al. (2022) [206]	AD-MSCs	Refractory Systemic Lupus Erythematosus	AD-MSC transplantation was associated with favorable safety and was efficient in reducing urine protein excretion and disease activity, single dose may not be adequate for long term remission
Wang et al. (2013) [204]	MSCs	Refractory Systemic Lupus Erythematosus	28–32.5% of patients in complete clinical remission in 1 year, rising to near 50% among those observed 4 years later
Sun et al. (2018) [209]	UC-MSCs	Kidney Transplant	UC-MSCs could achieve substantial reduction in the incidence of delayed graft function but there was no significant difference between MSC and non-MSC groups
Tan et al. (2012) [208]	MSCs	Kidney transplant	The use of autologous MSCs compared with anti-IL-2 receptor antibody induction therapy resulted in lower incidence of acute rejection, decreased risk of opportunistic infection, and better estimated renal function at 1 year

**Table 7 ijms-26-09659-t007:** Summary of selected studies on stem cell applications in neurology. This table highlights the types of stem cells investigated, the neurological disorders targeted, and the reported outcomes. Abbreviations: MSCs: Mesenchymal Stem Cells, HSCs: Hematopoietic Stem Cells, iPSCs: induced Pluripotent Stem Cells, NSCs: Neural Stem Cells, ALS: Amyotrophic Lateral Sclerosis, HD: Huntington’s Disease, MS: Multiple Sclerosis, PD: Parkinson’s Disease, AD: Alzheimer’s Disease, TBI: Traumatic Brain Injury, SCA: Spinocerebellar Ataxia, PSP: Progressive Supranuclear Palsy, SCI: Spinal Cord Injury.

Study	Stem Cell Type	Neurological Disorder	Outcome
Cecerska-Heryć et al. (2023) [217]	MSCs, HSCs, iPSCs	ALS, HD, MS, PD	Slowed ALS progression, reduced Htt aggregation in HD, immune recalibration in MS, accurate PD modeling
Namiot et al. (2022) [212]	MSCs	Brain injuries, stroke, MS, brain tumors	Dominance of MSCs in trials, most trials in early phases
Pappolla et al. (2024) [233]	NSCs, MSCs	Stroke, MS, ALS, TBI, PD, AD	Translational difficulties, potential of stem cell-derived exosomes
Khandia et al. (2024) [235]	Various stem cells	AD, PD, ALS, HD, SCA, PSP	Improved synaptic plasticity, apoptosis inhibition, reduction in tau-phosphorylation and Aβ production in AD
Isaković et al. (2023) [236]	MSCs	PD, AD, ischemic stroke, glioblastoma, MS	Safety established, concerns about immunocompatibility and tumorigenicity
Zayed et al. (2022) [237]	Various stem cells	AD, PD, HD, MS, ALS	Tailoring stem cells to specific disease defects, mixed clinical trial results
Izrael et al. (2025) [238]	Pluripotent stem cells	SCI, PD, ALS, retinal diseases	Preclinical and clinical trial progress, regulatory approvals
Ying et al. (2023) [239]	Various stem cells	CNS diseases	Cell replacement, immunoregulation, neurotrophic effects
Zhang et al. (2024) [240]	MSCs	Congenital nervous system and neurodegenerative diseases	Potential in treating neurological diseases, preclinical and clinical evidence
Liu et al. (2024) [241]	Various stem cells	Ischemic brain injury	Promising preclinical and clinical results, various administration routes
Yang et al. (2024) [234]	NSCs	Neurological diseases	Neuroprotection, axonal regeneration, remyelination
Wei et al. (2023) [242]	Various stem cells	Neurodegenerative diseases	Combination with nanotechnology to enhance efficacy

**Table 8 ijms-26-09659-t008:** Summary of stem cell therapy applications in hematologic diseases. This table outlines the type of stem cell therapy used for various blood disorders and highlights associated clinical outcomes. Abbreviations: HSCT, hematopoietic stem cell transplantation; AML, acute myeloid leukemia; ALL, acute lymphoblastic leukemia; SCD, sickle cell disease; GVHD, graft-versus-host disease; MSCs, mesenchymal stem cells.

Disease/Condition	Stem Cell Therapy Utilized	Outcomes
Acute Myeloid Leukemia (AML) [294,295]	Allogeneic Hematopoietic Stem Cell Transplantation (HSCT)	Increased survival, potential cure, but with risks of graft-versus-host disease (GVHD) and relapse
Acute Lymphoblastic Leukemia (ALL) [296]	Allogeneic HSCT	Improved survival rates, potential cure, similar risks as AML
Thalassemia [297]	Allogeneic HSCT, Autologous HSCT with gene therapy	Curative potential, reduced transfusion dependency, risks include GVHD and graft rejection
Sickle Cell Disease (SCD) [297]	Allogeneic HSCT, Autologous HSCT with gene therapy	Curative potential, reduced vaso-occlusive crises, risks include GVHD and graft rejection
Graft-versus-Host Disease (GVHD) [298,299]	Mesenchymal Stem Cells (MSCs)	Reduction in GVHD severity, improved engraftment
Hematopoietic Support Post-Transplant [299]	MSCs	Enhanced hematopoiesis, reduced tissue toxicities
Non-Malignant Hematologic Disorders (e.g., Aplastic Anemia) [300]	Allogeneic HSCT	Potential cure, improved hematopoiesis, risks include GVHD and infections

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
