# Peer review of "Utilization of Stem Cells in Medicine: A Narrative Review"

_ijms, 2025, doi:10.3390/ijms26199659_

Round 1
Reviewer 1 Report
Comments and Suggestions for Authors
- Abstract and Introduction should clearly define the scope of the review and highlight what makes this paper unique or timely in comparison to previous review papers.
- In section 3, each sub-section explores stem cell applications in different disorders. While each provides useful information, the review would benefit from a concise, integrative discussion that links them together.
- Consider including ethical or regulatory limitations with the application of stem cells as therapy.
- Avoid redundancy and combine overlapping content. For instance, details about lupus/SLE are found in both sections 3.2.6 and 3.8.2.
- Consider the reference format and maintain consistency.
- A conclusion and future perspective section should be added to summarize the main findings or conclusions of the review, along with any limitations and challenges, highlighting existing gaps or recent developments.
- Consider including some figures/schematic illustrations in the manuscript, which make the review more interesting to readers.
- There are a few minor wording issues as well.
In lines 21-22, the phrase “The terminology encompasses a broad spectrum" is unclear. If the goal is to refer to stem cells, it would be clearer to say something like: "The term ‘stem cell’ encompasses...".
Also, in lines 23-24, consider listing some of the “numerous medical fields” discussed in the review.
The phrase “providing a detailed discussion” in lines 25-27 is too vague and could be replaced with a more specific closing statement that highlights the paper’s main contributions.
Author Response
Please find it attached

Reviewer 2 Report
Comments and Suggestions for Authors
The manuscript titled “Utilization of Stem Cells in Medicine: A Narrative Review” provides a comprehensive overview of the current applications of stem cells across various medical specialties. The authors have invested considerable effort in compiling recent literature, including clinical trials and meta-analyses up to 2025, while presenting both the therapeutic potential and challenges of stem cell-based interventions.
To improve the clarity, depth, and scientific rigor of the manuscript, I recommend the following revisions:
-Although the review is narrative, it would be helpful to include a brief explanation of the literature selection process. This should cover the databases that were consulted, the keywords that were used, and any inclusion or exclusion criteria that were applied. Including this information would enable readers to better understand the scope and reliability of the cited sources.
-A dedicated section that compares the primary types of stem cells (e.g., MSCs, iPSCs, ESCs) in terms of clinical efficacy, safety profiles, and translational potential would provide valuable context and help readers navigate the complexities of the field.
-The manuscript contains several sections with lengthy lists of studies that lack adequate synthesis. I recommend condensing these sections and emphasizing key findings, emerging trends, and areas of controversy. This approach would enhance readability and establish a clearer narrative throughout the document.
-The conclusion can be strengthened by summarizing the most promising clinical applications, acknowledging current limitations, and outlining specific directions for future research and clinical translation.
-A thorough language revision is advised to enhance fluency and eliminate redundancy. This would make the manuscript more accessible and engaging, especially for readers from diverse disciplinary backgrounds.
In particular:
- Include a brief paragraph that outlines the literature search strategy, detailing the databases used, the time frame covered, and the specific keywords employed.
- Provide a comparative table or section that summarizes the characteristics and clinical relevance of various types of stem cells.
- Simplify the study summaries by concentrating on key interpretive insights instead of offering extensive listings.
- Introduce a discussion that examines shared challenges, such as immunogenicity, tumorigenicity, and ethical concerns.
- Revise the conclusion to present a more focused synthesis and offer a forward-looking perspective.
- Consider professional language editing to enhance clarity and improve the overall flow of the document.
Comments on the Quality of English LanguageA thorough language revision is advised to enhance fluency and eliminate redundancy. This would make the manuscript more accessible and engaging, especially for readers from diverse disciplinary backgrounds.
Author Response
Please find it attached

Round 2
Reviewer 1 Report
Comments and Suggestions for Authors
My comments have been addressed